# National Profile of Caregivers’ Perspectives on Autism Spectrum Disorder Screening and Care in Primary Health Care: The Need for Autism Medical Home

**DOI:** 10.3390/ijerph182413043

**Published:** 2021-12-10

**Authors:** Sarah H. Al-Mazidi, Laila Y. Al-Ayadhi

**Affiliations:** 1Department of Physiology, Faculty of Medicine, Imam Mohammed Ibn Saud Islamic University, P.O. Box 5701, Riyadh 11432, Saudi Arabia; 2Department of Physiology, Faculty of Medicine, King Saud University, P.O. Box 2925, Riyadh 11461, Saudi Arabia; lyayadhi@ksu.edu.sa; 3Autism Research and Treatment Centre, King Saud University, P.O. Box 2925, Riyadh 11461, Saudi Arabia

**Keywords:** autism spectrum disorder, medical homes, autism, primary healthcare

## Abstract

Although autism spectrum disorder (ASD) is a common developmental disorder, primary healthcare providers show a deficit in providing early diagnosis. To understand parents’ experience and perspective in the diagnosis and intervention process of their children, a survey was deployed through social media to parents’ with at least one child diagnosed with ASD. The survey included parents experience, satisfaction and perception in the diagnosis process and services provided for their children, stigma and type of support received. A total of 223 participants were enrolled. Although 62% of ASD patients were diagnosed by three years old, most diagnoses (66%) were non-physician initiated. Additionally, 40.8% of the parents reported that the services required for their child are available in their area of residence, but only 7.9% were satisfied with these services. Parents who received psychological support (9.9%) started early intervention, and their children have a better prognosis (*p* ≤ 0.005). Stigmatized parents were more likely to delay intervention (*p* ≤ 0.005). Parents’ perception is to have qualified healthcare and educational professionals experienced in ASD. Our findings suggest that a specialized family-centred medical home for ASD patients would significantly benefit ASD patients, increase parents’ satisfaction, reduce parents’ stress, and ease their children’s transition to adolescents.

## 1. Introduction

Autism spectrum disorder (ASD) is a cluster of neurodevelopmental disorders which are characterized by restricted and repetitive behavioural patterns, and limitations in social interaction and communication [1]. Children with ASD have other comorbidities such as gastrointestinal problems, food and skin allergies, and epilepsy that needs direct medical attention [2]. According to the Centre for Disease Control and Prevention, the incidence of ASD has been estimated to be 1 in 54 children in the United States. The reason for this increase is not clear; it might be due to the changes in the diagnostic criteria or increased awareness of ASD among the community, medical professionals and educators [3,4]. Early diagnosis at three years old is crucial and significantly improves ASD patients’ social behaviour and cognitive skills [5,6]. 

Since ASD has increased in the past decade, developing a special care system that provides optimal care for the children and their parents is essential. Although it is a common developmental disorder, primary health care providers show a deficit in providing early diagnosis. Physicians at the primary health care have a vital role in the early detection of ASD, which ensures optimal improvement for the children and reduces stress on caregivers [7]. 

Many diagnostic instruments are available for ASD, which includes validated scales, subscales, and checklists [8]. Ideally, diagnosis of ASD requires a multidisciplinary team of professionals specializing in the assessment of ASD [9]. This team includes a paediatrician, child psychiatrist, psychologist, speech-language pathologist, occupational and behavioural therapist [10]. Access to centres with a multidisciplinary team or experienced physicians in ASD is difficult and unavailable in all areas [11].

The American Academy of Pediatrics established special policies and standards for what is known as medical homes. A medical home is a specialized medical centre that is an accessible, coordinated and compassionate family-centred primary health care centre [12,13]. A medical home creates a partnership between caregivers and their paediatrician, clinicians, early childhood professionals, and educators, all of whom are specialized in ASD intervention [14]. Medical homes are cost effective, improve providers’ and patients’ satisfaction, reduce the psychological and financial burden on caregivers, and children are more likely to receive high-quality care [12,13,15]. Medical homes also provide healthcare providers trained to address medical and mental needs for ASD patients transitioning into adolescence and adulthood [16]. 

The aim of our study is to understand parents’ experience in the diagnosis and intervention process of their children and their perspective of the best care for ASD children and their parents. 

## 2. Methods

### 2.1. Study Design

We conducted a non-experimental, cross-sectional study of parents’ satisfaction and perception of healthcare services for ASD patients. The study was approved by the Institutional Review Board. 

### 2.2. Study Setting

This study sought to understand the parental experience of healthcare services provided for their ASD children in Saudi Arabia. Parents from different cities participated in this survey. The survey was distributed electronically to the families that have at least one child diagnosed with ASD through physicians, autism schools, autism centres, psychologists and educators using social media applications (Twitter, WhatsApp, and LinkedIn). 

### 2.3. Participants and Procedures

Participant characteristics: Participants consisted of 224 parents of one or more children (males and females) with ASD diagnosed and treated in Saudi Arabia either by a physician or other healthcare provider (such as a psychologist). 

The survey and consent: The survey had an introductory statement that describes the study’s aims and that their participation was voluntary with complete anonymity ensured. It also stated that they were allowed to withdraw from the study at any time. 

A 42 item self-administered online survey using Google Forms**^®^** (Google LLC, Mountain View, CA, USA) was prepared and administered via Google forms to the parents of ASD children. The survey was available from October 2020 to June 2021. 

Six participants were invited to pilot-test the initial draft survey to validate it; minor modifications were made based on their feedback. Then, the survey was distributed using social media platforms, including Twitter, WhatsApp, and LinkedIn. The survey was sent three times to each potential participant to maximize our response rate.

### 2.4. Measures

The survey included both open-ended and closed-ended questions which evaluated the following: 

#### 2.4.1. Diagnosis Process and Referral Source

We asked the parents about the details that were experienced during the diagnosis and referral process of their children using closed-ended questions as follows: age of the child when diagnosed with ASD, the duration between the first visit to the healthcare professional and time they received the diagnosis, the healthcare professional that initiated the diagnosis, the main symptom that led to the diagnosis, the type of clinic that diagnosed their child, and the availability of the healthcare services in their area of residence to diagnose their child.

#### 2.4.2. Parents’ Satisfaction with the Service Provided for ASD Patients

Parents evaluated the health care services provided for their children according to the following: availability of the service in their area of residence, type of intervention received and treatment plan, emotional and psychological support received, subjective improvement after intervention as rated by the parents, type of follow-up clinic, early intervention (at the age of three years), yearly cost of their child intervention. 

These questions were closed-ended, then an open-ended question was provided to explain the reasons for their dissatisfaction with the healthcare provided for their child. The open-ended question was qualitatively analysed using thematic framework analysis 

#### 2.4.3. Stigma and Stress Level of Parents

Parents rated their stress level during the diagnosis and intervention process of their children. All items were rated on a 5-point Likert scale, ranging from never (zero) to always (five), where higher scores indicated greater stress. We asked the parents if they are stigmatized with their child’s diagnosis. Answers were provided as (Yes/No).

#### 2.4.4. Parents’ Perception of the Services That Should Be Provided for ASD Patients

To achieve patients’ and parents’ satisfaction with the available healthcare services, the parents were provided an open-ended question that explains their perspective on services that should be provided according to their point of view. This question was qualitatively analysed using thematic framework analysis.

### 2.5. Statistical Analysis

All the statistical tests were performed using the Statistical Package for Social Science (SPSS) software version 26. Descriptive statistical analysis was used to analyse the items included in the survey such as participants’ demographics and other study outcomes. Responses were presented as frequencies and percentages. Chi-square and Fisher’s exact tests were used to compare responses between variables in different categorical measures. *p* values that are equal to or less than 0.05 were considered significant. In the present study, saturation of data is the point at which no new themes were created [17]. The hybrid structure of our survey, with closed and open-ended questions, provided flexibility to apply both quantitative and qualitative analytic methods for analysis. After gathering the results, the quantitative data was tabulated, and the textual responses were collated for further qualitative analysis using a thematic analysis approach to identify themes and enable us to understand participants perspectives of health care services provided for their children.

## 3. Results

The survey was available for nine months (October 2020 to June 2021). Within the thematic analysis, we examined the recurrence of certain issues in the responses, that included (i) diagnosis, (ii) intervention, and (iii) quality of service. After reaching 200 participants, there were no new themes generated. Therefore, it was deemed that the data collection had reached a saturation point. We continued data collection for 24 more participants to ensure and confirm that no new themes are emerging. 

### 3.1. Participant’s Demographics

#### Survey Sample

A total of 224 participants (76.7% males) responded to the survey and completed a set of questions to screen for eligibility. Data saturation was reached and only one participant was excluded. Study demographic data are shown in Table 1. Male participants were significantly more than females. Participants’ financial status was normally distributed. Table 2 shows family history. Most families have at least one working parent. The most reported disabilities of ASD siblings were learning disabilities (22.9%) and delayed speech (22.4%). The most reported diseases in the family were asthma (17%) and Rheumatoid Arthritis (10%).

### 3.2. Survey Measures

#### 3.2.1. Diagnosis Process and Referral Source

Descriptive information on the diagnosis process and referral source are shown in Table 3. The most common ASD symptoms reported within the children were delayed speech (70.9%), difficulties in communication with the child (61.9%), repetitive movements (50.7%) and abnormal playing behaviour (48%). 

According to the parents, only 32% were initially diagnosed by a paediatrician, while 46% were diagnosed by a psychologist and 22% by a speech therapist (Figure 1). About 64% of the ASD children received their diagnosis by the age of three-years-old, and 61% of them were diagnosed in a governmental medical facility. Only 40.8% of the parents reported that the services provided for their children were available in their area of residence. Additionally, many of the parents reported that the diagnosis process was long and costly (Table 3).

#### 3.2.2. Parents’ Satisfaction with the Service Provided for ASD Patients

Closed-ended questions were analysed to determine parents’ satisfaction with the provided healthcare services. Then, an open-ended question was analysed using framework analysis, which led to the identification of categories within our themes as follows: (i) wrong initial diagnosis, (ii) availability of qualified healthcare professionals, and (iii) settings of healthcare centres for ASD. 

The response to all themes was similar, but the frequency of participants with dissatisfaction with the settings of healthcare centres for ASD was found to be the greatest. For example, parents that reported the availability of services in their area of residence responded to the open question as follows:-*There are many centres available for autism, but they are not as qualified as the centres in a neighbouring country.*-*Unfortunately, there is no accurate diagnosis for our children, and the first diagnosis for my child was wrong.*-*Unfortunately, there are no qualified healthcare professionals for the treatment and diagnosis of autism so far.*

Early intervention was significantly correlated with service availability (*p* ≤ 0.05) and child improvement (*p* ≤ 0.03). Table 3 shows that 72.6% of parents who reported that their child improved have started early intervention. Interestingly, most parents who started the intervention early work either in the health or education sectors (29.1%) compared to other parents (*p* ≤ 0.01). There are 27% of parents who did not start intervention early. None of these parents work in the health sector (Table 2). 

According to the parents point of view, which was subjectively based on their expectation of intervention, the most effective intervention was behavioural therapy (64.8%) followed by learning-directed therapy (52.4%). Other interventions include speech therapy (36.7%) and occupational therapy (41%) (Figure 2). Although 40.8% of the parents stated that the treatment and diagnostic services are available, only 7.9% were satisfied with the provided services. 

We found that psychological support to the parents is crucial for their children’s improvement. Only 9.9% of the parents received proper psychological support during their child’s diagnosis and intervention process in the current study. 

Parents who received psychological support were more likely to start early intervention (84%) than those who did not receive any psychological support (*p* ≤ 0.005). Additionally, all parents who received psychological support reported that their children had improved, while 25.6% of parents who did not receive psychological support reported that their children had improved with the intervention (*p* ≤ 0.001). 

#### 3.2.3. Stigma and Stress Level of Parents

The long duration until the official diagnosis was made had increased stress levels on the parents; for example, of all the 45.5% of total parents who reported extreme stress (Figure 3), 19.8% of them received the diagnosis within a month, while 40.6% of them received it after one year. 

Although most of our participants are not stigmatized by their child’s diagnosis, we found that stigmatized parents did not start treating their children once they were diagnosed (*p* ≤ 0.005). 

#### 3.2.4. Parents’ Perspective on the Services That Should Be Provided for ASD Patients

The parents view and experiences and their perspectives on healthcare services for ASD children are similar in all cities in Saudi Arabia. There are three main categories within our themes analysed using framework analysis, and parents’ responses were similar in each category. 

We chose some of the parent’s point of view as follows: -View on diagnosis:


*We hope to have excellent centers for children with mild autism and establish special schools to have curricula specialized in their way of thinking. I hope that autistic children get great attention in education, diagnosis, and training, because we, as parents, have been confused, and no guidance was provided to us, and access to services is still difficult for most of us.*


*We need the diagnosis to be immediate, accurate, with a precise treatment plan prepared by a specialized team, according to the needs of each child and immediately refer to an appropriate center, according to the specialists’ point of view and not letting the parents struggle due to their lack of knowledge. Therefore, parents should be directed to the appropriate place that provides necessary services for their children*.


*We should emulate the experience of other countries and transfer knowledge and bring in the best experts in this field.*


*The difference of opinions during diagnosis has caused us great humiliation*.

*There should be an obligatory routine screening for autism in the preschool years*. 

-View on intervention 

*To have a complete center with all required medical, educational, social, and psychological consultants for parents and children. A center that would also prepare them for adulthood and merge smoothly into society and have jobs in the future*. 

*To increase the governmental centers because the private centers cause a great financial burden on us*. 

*Provide complete healthcare services that are accessible to all of the autism community*.

-View on parents’ support

*In my opinion, the most important thing is the psychological stability of the parents*.

*We suffer from our preoccupation with our jobs for long periods, as we cannot take care of our autistic children as we should, and there is no safe place that accepts their presence for full working periods*.


*Increase awareness of autism and have special entertainment and sports centers for them.*


## 4. Discussion

Whether there is an actual increase in the prevalence, or it is due to the awareness of ASD in the community, it is crucial to evaluate the current services offered to ASD patients and their parents to provide optimal care and increase their satisfaction with the offered services. This study aimed to evaluate the current healthcare services in the diagnosis and intervention of ASD and understand parents’ perspectives on healthcare offered for them and their children. In our study, most parents reported that the services are unavailable in their area of residence or they are not aware of the available services. We also found that all parents working in the health field are treating their children, which indicates these healthcare services are not adequately promoted to the public. Moreover, most parents were not satisfied with the available services and reported that the diagnosis process was long and costly. The main reason for the high cost of ASD intervention is because it involves many specialized medical and educational services. One suggested solution to the high cost of intervention is to promote health care volunteering services at public clinics [18]. 

Parents of an ASD child experience chronic stress, especially during the diagnosis process [19,20]. Our results showed that psychological support is essential for ASD parents. Parents reported that the availability of high-quality services for their children might reduce the amount of stress they are experiencing. This finding is consistent with the results reported by Ault et al., who reported that ASD parents experience more stress and poor mental health when services are unavailable [21]. 

Prolonged duration until diagnosis and service availably are crucial to reduce parents’ stress. In our study, some parents had multiple visits to the primary health care physician and paediatrician; it took as long as two years to have an official diagnosis and start intervention. This might be because most physicians in primary health care are not experienced in diagnosing ASD. This finding was supported by previous studies that reported that ASD is difficult to define, diagnose, and the physicians cannot translate their theoretical knowledge of ASD into their clinical practice, which shows a lack of experience in ASD diagnosis in primary health care physicians [22,23]. To overcome this, a previous study suggested a valid scale easily performed by any healthcare provider and does not require special training which might be distributed to primary health care and even to educators [5]. 

Early intervention has a significant impact on ASD patient’s progress and improvement [24,25]. Although The American Academy of Pediatrics recommended routine screening of ASD for children aged from 18 months to 24 months, few centres or even countries follow this screening routine [26]. Early years educators can also be trained to use a simplified scale and report a possible ASD child. Teachers should also be trained to handle students with ASD and co-existing behaviours such as restrictive and repetitive behaviours and anxiety [27,28,29]. Although certification in special education is an advantage for ASD education, a study showed that special certification is not necessary for optimal academic outcomes in ASD students [30]. This is beneficial for some ASD children studying in regular schools or even merging some ASD children into the regular teaching system.

In our study, most parents were not stigmatized by their ASD child, and we found no correlation between stigma and service availability, income, or type of psychological support to the parents, and it did not affect their child’s treatment or education. A multicentre study reported that stigmatized parents need psychological support and that stigma increases with lower income and services availability [31]. Another study reported that parents who feel psychologically supported would positively impact their children’s care, which was also a finding in our study [32]. This requires proactive action to develop a special psychological support service to help parents with their children’s intervention.

Knowing the parents’ experience is vital to evaluating our current diagnostic and intervention services and is the backbone of developing a satisfying healthcare system for parents, ASD patients and health care workers [33]. In our study, we asked for the parents’ perspectives for ASD health care, which can be summarized as the following: there is a need for qualified paediatricians and primary care physicians to diagnose ASD and manage co-existing disorders such as gastrointestinal problems, seizures, allergies and to refer them to manage any behavioural or psychiatric disorders that co-occur in ASD such as attention-deficit hyperactivity disorder, anxiety, obsessive-compulsive disorder, and mood disorder. They also need a high-quality centre that meets the international standards in ASD intervention that includes psychologists, speech therapists, occupational therapists, and all medical tools and consults needed for their children that is accessible and affordable to all families. They also requested a centre that helps their children’s transition into adolescence and adulthood and provides families with the necessary support. 

All these suggestions fit into what is known as medical homes. A medical home is based on a partnership between physicians and parents. It is a society-based and coordinated primary healthcare centre that includes all care services that an ASD patient might need, such as qualified physicians, medical personnel, early educators and family support. All services and staff at a medical home are specifically trained to deal with ASD patients and co-existing medical and mental disorders and improve child and family diagnostic and intervention experience [34]. 

Recently, there is a global shift toward medical homes for ASD. It is cost-effective and relieves financial and psychological burdens on working parents. In our study, most parents reported that the services provided to their children were not satisfying, and many reported that they do not have access to healthcare services specific to ASD in their area. Our findings are consistent with a national survey in the United States, which found that ASD patients who do not have access to a medical home have higher burdens in all study indicators than those in a medical home [12]. Additionally, a study reported that parents of ASD children in medical homes are less stressed than the other ASD parents and reported improvement in their mental health [15]. Families are not aware of the concept of a medical home; thus, they lowered their expectation of healthcare services and did not expect to receive comprehensive care, and their treatment goals are limited to maintaining their children’s current health [32,35]. 

In our study, most families reported that the diagnosis was non-physician initiated, which led to delayed diagnosis. The reason might be because the parents are more aware and educated, which makes them seek other health care professionals; as in our study, most parents reported that the primary source of information during the diagnostic process was gained from the internet. The difference in referral process in ASD was previously reported, which suggests that a medical home model may reduce the referral duration and reduce the chances that ASD patients are diagnosed as developmental delay cases [7]. In our study, parents were stressed during the diagnosis process, which would be significantly reduced in a family-based centre, as reported previously by Myers et al. [20]. 

## 5. Strengths and Limitations

Strengths of this study included a geographically diverse sample of parents and adequate sample size, a saturation of qualitative data, and the use of technology to recruit the participants, which decrease selection bias. Study limitations include the following: survey questions relied on parent self-report associated with recall bias. Future studies with larger samples size are needed to explore the relationship between types of health care services during the diagnostic and intervention process for ASD. Data collection was terminated because sample adequacy and saturation were reached, and no new insights were found during data analysis. 

## 6. Conclusions

In this study, we used self-report measures to understand the parent experience during the diagnostic and treatment process and to seek their perspective in ASD healthcare services in their area of residence. Our findings suggest that a specialized family-centred medical home specific for ASD patients might have a significant impact in increasing parents’ satisfaction with healthcare, reducing parents’ stress, and easing the transition of their children to adulthood. 

## Figures and Tables

**Figure 1 ijerph-18-13043-f001:**
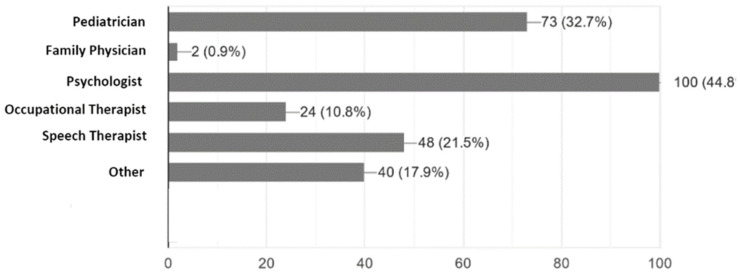
Healthcare professionals that initiated ASD diagnosis according to caregivers experience.

**Figure 2 ijerph-18-13043-f002:**
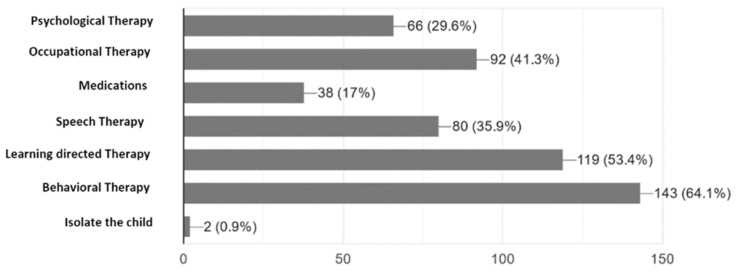
Caregivers subjective point of view on the most effective intervention for their children.

**Figure 3 ijerph-18-13043-f003:**
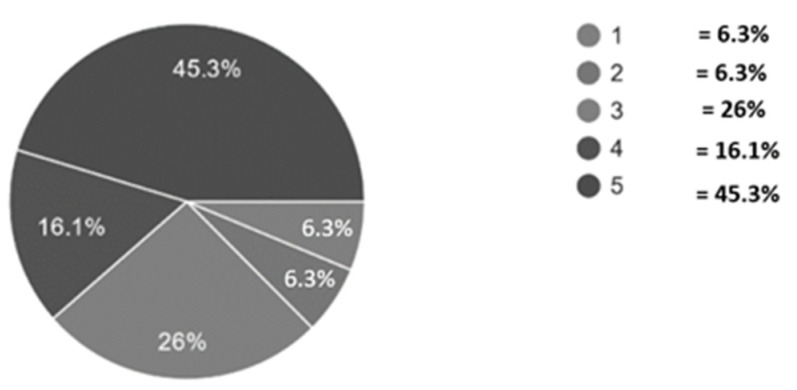
Stress level experienced by ASD caregivers. Level of stress indicated as 1 = Never and 5 = very often where higher score correlates to more stress.

**Table 1 ijerph-18-13043-t001:** Demographics.

Characteristics	Variable	Percentage
Child Age (years)	1–3	1.3%
	4–6	13.9%
	7–9	20.7%
	10–12	35.4%
	13–15	18.8%
	More than 16	9.9%
Sex	Male	76.7%
	Female	23.3%
Area	Riyadh area	38.6%
	Dammam area	18.4%
	Abha and Jizan	2.3%
	Makkah area	16.2%
	Al-Khafji	1%
	Hail	1%
	Qassim	2.7%
	Madina	5.8%
	Al-Ehsaa	6.7%
Monthly Income	Low (less than 1500 $) SAR	27.8%
	Middle low (1500–2500 $)	24.3%
	Middle high (2600–4000 $)	25.1%
	High (more than 4000 $) SAR	22.2%
Property	Owned	53.4%
	Rented	46.6%

**Table 2 ijerph-18-13043-t002:** Family History.

Characteristics	Variable	Percentage
Parents Related	Yes	38.6%
	No	61.4
Mothers Age	20–29	8.1%
	30–39	42.6%
	40–49	32.3%
	50–59	17%
Fathers Age	20–29	1.8%
	30–39	26%
	40–49	36.3%
	50–59	28.3%
	60+	7.6%
Mothers Education	Higher Education	3.9%
	University	47.1%
	Diploma	5.8%
	High school	43%
Fathers Education	Higher Education	8.4%
	University	32.7%
	Diploma	16.1%
	High school	41.7%
Work field Mother	Education	27.4%
	Health	6.7%
	Administrative	3.6%
	Self-Employed	5%
	Housewife	57.4%
Work field Father	Education	15.7%
	Health	1.8%
	Engineering	5%
	Administrative	17.5%
	Self-Employed	12.6%
	Unemployed	10.4%
	Other	35.4%
Siblings with other disorders	Developmental delay	4.9%
	Learning disabilities	22.9%
	Mental Disorder	8.5%
	Delayed Speech	22.4%
Parents with disorders	Delayed Speech	2.7%
	Learning Disabilities	2.7%
	Depression	2.3%
	Bipolar	1.3%
	Other psychological Disorders	4.9%
Family members disorders	Rheumatoid Arthritis	10.8%
	Multiple sclerosis	6.7%
	Alzheimer	4.9%
	Parkinson’s Disease	2.7%
	Asthma	17.9%
	Psoriasis	3.1%
	Autoimmune Disease	4%

**Table 3 ijerph-18-13043-t003:** Diagnosis and intervention.

Characteristics	Variable	Percentage
Childe age when diagnosed	1–3	64.1%
	4–6	32.3%
	7–9	3.5%
Child Education	Private School	13%
	Governmental School	17%
	Special needs school	46.6%
	Not at school Because:	
	Young Age	4.5%
	School not available	12.6%
	Other	6.3%
How long did it take to be diagnosed?	1 Month	23.5%
	1–3 Months	20.2%
	4–7 Months	11.7%
	8–12 Months	24.6%
	More than A year	20%
Type of clinic that made the diagnosis	Government	59.2%
	Private	40.8%
Service Available for diagnosis and treatment?	Yes	40.8%
	No	40.4%
	I don’t know	18.8%
Once diagnosed, did he/she start treatment?	Yes	72.6%
	No	27.4%
Did the Childe improve after treatment?	Yes	74.9%
	No	25.1%
Did you receive the appropriate psychological support?	Yes	9.9%
	No	40.4%
	To some extent	49.7%
Satisfaction of services provided	Yes	7.2%
	No	49.8%
	To some extent	43%
Stigma	No	73.1%
	Yes	26.9%
Cost of Diagnosis	None	28.7%
	Up to 1000 SR	17%
	Up to 3000 SR	15.2%
	Up to 6000 SR	15.7%
	Up to 10,000 SR	23.3%
The primary symptom noticed	Speech delay	70.9%
	Communication issues	61.9%
	Repetitive movements	50.7%
	Different playing style	48%
	No danger senses	38.6%
	Socially isolated	35.4%
	Annoyed by sounds or colors	26%
	Learning problems	25.1%

## Data Availability

Data in this research are available in tables section of this manuscript.

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
