# Peer review of "National Profile of Caregivers’ Perspectives on Autism Spectrum Disorder Screening and Care in Primary Health Care: The Need for Autism Medical Home"

_ijerph, 2021, doi:10.3390/ijerph182413043_

Round 1
Reviewer 1 Report
Dear authors. I read with interest your paper on parental satisfaction and perception of their children's diagnosis and intervention processes. I think it's an important topic and that your manuscript does a good job addressing this topic. You bring up important points and I hope my comments below will help you improve the presentation and clarity of these points:
I had multiple comments about English and grammar which I detail in the attached file, along with several other minor comments and suggestions in this regard. I highly recommend that a professional editor will go over the paper, especially the abstract, intro and discussion and allow your findings and ideas to be effectively described and make their impact on readers.
In addition, I have the following comments I recommend you will address:
1)measures: for some of the measures taken, like stress, for example, there are numerous standardized and well-known questionnaires to measure them. Were any of those used? If not, who came up with the questions? what were they? I also suggest attaching the questionnaire as an appendix.
2)It is helpful and even necessary to understand the standards of diagnosis in the country. who is considered a standard diagnosis provider? What is the "gold standard" of diagnosis? This information should be provided early (in the intro) to allow the readers to interpret the results presented in this regard.
3) in the results, there is an entire section devoted to parents' subjective opinions about their children's improvement and treatment efficacy. While this is perhaps interesting/relevant, the parents are not professionals so their subjective opinions about these matters are not a scientific source for recommendations, nor are they related to the research question, in my opinion.
4)presentation of the conclusions: The authors must be extra careful in how they present their findings. I urge the authors to ensure not to argue for causality where they did not prove for one. One example is in the sentence: "Our results showed that psychological support is essential to reduce parents' stress and optimize their children's improvement". --> I argue that this could only be shown by having two groups of parents - one with and one without psych support and measuring their stress levels before and after and comparing them. Since the authors merely asked retrospectively about stress and compare without intervening, all they can argue is that parents with psych support reported less stress in hindsight (or something that more accurately describes the actual findings). This is a critical point.
critical comment: either I missed it or I have not received the tables and figures, thus was unable to comment on them.

Reviewer 2 Report
Dear authors:
Thank you for the work that addresses an issue of interest to individuals and families of children with autism. Just a few recommendations:
- The method should be improved by incorporating more clearly what is the design of the instruments and what items are included. Also, its validity could be discussed.
- The characteristics of the sample should be included as part of the method.
- The wording of the article should be revised (spaces, commas, capital letters...).
